# Analysis of Publication Activity and Research Trends in the Field of AI Medical Applications: Network Approach

**DOI:** 10.3390/ijerph20075335

**Published:** 2023-03-30

**Authors:** Oleg E. Karpov, Elena N. Pitsik, Semen A. Kurkin, Vladimir A. Maksimenko, Alexander V. Gusev, Natali N. Shusharina, Alexander E. Hramov

**Affiliations:** 1National Medical and Surgical Center Named after N. I. Pirogov, Ministry of Healthcare of the Russian Federation, 105203 Moscow, Russia; 2Baltic Center for Neurotechnology and Artificial Intelligence, Immanuel Kant Baltic Federal University, 236041 Kaliningrad, Russia; epitsik@kantiana.ru (E.N.P.); skurkin@kantiana.ru (S.A.K.); vmaksimenko@kantiana.ru (V.A.M.); nshusharina@kantiana.ru (N.N.S.); 3K-Skai LLC, 185031 Petrozavodsk, Russia; 4Federal Research Institute for Health Organization and Informatics, 127254 Moscow, Russia

**Keywords:** artificial intelligence, medical data, medical area, unsupervised learning, supervised learning, deep learning, artificial neural network, machine learning

## Abstract

Artificial intelligence (AI) has revolutionized numerous industries, including medicine. In recent years, the integration of AI into medical practices has shown great promise in enhancing the accuracy and efficiency of diagnosing diseases, predicting patient outcomes, and personalizing treatment plans. This paper aims at the exploration of the AI-based medicine research using network approach and analysis of existing trends based on PubMed. Our findings are based on the results of PubMed search queries and analysis of the number of papers obtained by the different search queries. Our goal is to explore how are the AI-based methods used in healthcare research, which approaches and techniques are the most popular, and to discuss the potential reasoning behind the obtained results. Using analysis of the co-occurrence network constructed using VOSviewer software, we detected the main clusters of interest in AI-based healthcare research. Then, we proceeded with the thorough analysis of publication activity in various categories of medical AI research, including research on different AI-based methods applied to different types of medical data. We analyzed the results of query processing in the PubMed database over the past 5 years obtained via a specifically designed strategy for generating search queries based on the thorough selection of keywords from different categories of interest. We provide a comprehensive analysis of existing applications of AI-based methods to medical data of different modalities, including the context of various medical fields and specific diseases that carry the greatest danger to the human population.

## 1. Introduction

The expression ’medical technology’ is widely used to refer to a range of tools that can enable health professionals to provide patients and society with a better quality of life by providing early diagnosis, reducing treatment complications, optimising treatment and/or providing less invasive options, including reduced hospital admissions. Whereas before the Information Age, medical technology was mainly known for classical medical devices (e.g., implants, prostheses, stents, functional diagnostic systems, etc.), the development of information technology (IT) has led to a revolution in the development of specifically digital medical products and services, among which the greatest hope is the widespread adoption of artificial intelligence (AI) technologies.

Currently, the application of various IT solutions based on AI technologies is one of the most promising areas of digital healthcare transformation. The growing interest in AI is driven by several trends, including increasing hardware computing power, the development of cloud computing, the rapid accumulation of large digital biomedical datasets, and the development of machine learning (ML) algorithms. AI-based techniques, such as fuzzy expert systems, Bayesian networks, various classifiers, and artificial neural networks have long been used in various clinical situations in healthcare. According to analytics [1], the global healthcare AI market size was USD 8.19 billion in 2021, will grow to USD 10.11 billion in 2022 at a compound annual growth rate (CAGR) of 23.46%, and will increase to USD 49.10 billion at a CAGR of 48.44% by 2026.

We see AI as part of IT, capable of solving complex problems in areas where large structured marked-up datasets are accumulated, but without well-developed theory [2]. AI technology is effective where no clear rules, formulas, and algorithms can be specified to solve a problem, e.g., ‘Is there pathology on the lung X-ray image?’ ML technologies imply that instead of implementing some pre-formulated logical formula based on clear instructions, such as ‘if…— then…’, the algorithm is trained using a large amount of pre-prepared marked-up data and various mathematical methods that enable the computer program to identify that formula from empirical data and thereby learn to perform the task in the future, even in slightly different circumstances [3]. The Big Data approach uses the ML principles, including various classifiers, deep learning (DL), or pattern recognition, which in the case of medical applications involves training an intelligent system with repetitive algorithms to recognise what certain groups of symptoms or certain clinical (e.g., radiology, CT or MRI) images look like, i.e., to actually classify biomarkers of certain diseases [4].

Currently, the main areas of research and development in the field of medical AI technologies are diagnostics and prognosis of diseases and their complications [5], selection of personalised therapy [6,7], operation of personal medical assistants for real-time monitoring and assessment of patients [8], and the development of new drugs and support for their clinical trials [9,10,11]. A separate but still underdeveloped area is the development of robotic, truly autonomous devices for the healthcare sector [12,13].

Over the last decade, many AI-based algorithms have been approved in various countries and can, therefore, be introduced into clinical practice. Investment in AI for healthcare has been growing steadily since 2017 (Figure 1A). According to CB Insights, in 2021, total investment in companies offering various products based on AI technology was USD 12.2 billion (505 deals). By comparison, in 2020 this figure was USD 6.627 billion (397 deals), in 2019 it was USD 4.129 billion (367 deals), and in 2018 it was only USD 2.7 billion (264 deals). In addition, the temporal dynamics of the number of articles on artificial intelligence in medicine indexed in the PubMed database demonstrate a fast growing trend and has doubled in 2022 compared to 2015 (Figure 1B).

AI-based products can potentially improve the efficiency of medical doctors, nurses, and healthcare organisations by reducing the time needed to document the care process, ensuring patient routing and the necessary communication of all parties involved in the process [1,14]. The COVID-19 pandemic has significantly increased interest in the use of AI products, although, as noted in two recent meta-reviews [15,16], the effectiveness of AI-based models for predicting the severity of the COVID-19 disease has not been sufficiently high. Recent publications have shown that additional research and development, including the provision of independent clinical validation and cost-effectiveness assessments, are needed if AI systems are to be widely adopted in healthcare practice [17,18,19]. At the same time, the use of AI can create social and ethical problems related to security, privacy, and human rights [20].

One of the first developments of expert software in medicine is the MYCIN system [21], which was based on the fuzzy sets mathematical methods. It was created by Stanford University in the early 1970s to identify the causative agents of severe infectious diseases and calculate the required doses of antibiotics. We would now call it the clinical decision support system (CDSS) [22]. MYCIN CDSS was estimated to be 69% effective, in as many cases as the correct treatment was administered [21]. A CADIAG-2 expert system similar in mathematical principles based on fuzzy sets and fuzzy logic was created in the 1980s [23] at the University of Vienna and contained symptoms and diagnostic rules for 295 diseases, among them 185 rheumatic diseases (69 joint diseases, 12 spinal diseases, 38 soft tissue and connective tissue diseases, 45 cartilage and bone diseases, 21 regional pain syndrome) and 110 gastroenterological diseases (35 gall bladder and bile duct diseases, 10 diseases of the liver) [24,25].

Another AI-based software in medicine is IBM Watson [26]. Initially, this solution was aimed at an application in oncology to diagnose and recommend an effective treatment for each patient. To train IBM Watson, 30 billion medical images were analyzed, for which IBM had to buy Merge Healthcare. This demonstrates the importance to the development of AI in medicine of precisely access to marked and reliable medical data. It took the addition of 50 million anonymous electronic medical records, which IBM got its hands on when it bought the start-up Explorys. In 2014, IBM announced a partnership with Johnson & Johnson and pharmaceutical company Sanofi to train Watson to understand research and clinical trial results. This, the company claimed, would result in significantly shorter clinical trial times for new drugs and help doctors select therapies best suited to individual patients. Additionally, in 2014, IBM announced the development of Avicenna software, which can interpret both text and images. Different algorithms are used for each type of data. Avicenna will be able to understand medical images and records and will act as a radiological assistant. Another IBM project, Medical Sieve, has been working on a similar task. In this case, it is the development of an artificial intelligence ‘medical assistant’ that will be able to quickly analyze hundreds of images for abnormalities [27].

However, the use of CDSS is currently limited [28]. Despite all the benefits of CDSS that were widely reported in previous studies [29], certain factors that prevent their widespread use in healthcare. Some of them are social and psychological, such as negative perception of CDSS by clinicians [30] and lack of trust in the decisions made by CDSS [31], and the others are related to technical issues [32,33]. In addition, a significant research gap affecting the development of medical AI-based methods is a lack of appropriately labeled medical data [34]. This led to the development of various techniques of data labeling for AI-based methods [35,36].

So, there is an active adoption of AI technology in medicine at the moment. Medical AI systems exist in many forms, from the purely virtual (e.g., deep learning medical information management systems to assist doctors in making treatment decisions) to the cyberphysical (e.g., robots used to assist the treating surgeon) [37]. The ability of AI technologies to recognise complex associations and hidden structures using big data has enabled many healthcare diagnostic systems based on medical data of different modalities to perform as well as, and in some cases better than, physicians.

In this paper, we aim to explore the tendencies of using AI in medicine from an engineering and data science perspective. We identify and compare AI technologies for different data modalities used by researchers in different branches of medicine. We analyzed the results obtained from PubMed database using thoroughly constructed search queries based on keywords to obtain the most conclusive results with each query. We also used VOSviewer software to analyze the co-occurrence network and select five main clusters of keywords, each representing the research area that emerged in the scientific community over the last five years. We believe that the results of this research will increase understanding current directions of AI-based methods development in medical research, and will potentially help the specialists to determine the directions of their own studies.

## 2. Methods

### 2.1. Search Strategy

To search for articles, we utilized the database PubMed (https://pubmed.ncbi.nlm.nih.gov, accessed on 5 February 2023) using the queries of keywords. The methodology of the present review is to analyze the main research trends of artificial intelligence and machine learning applications in medicine. This task required the analysis of a large amount of paper, which excluded the possibility of the detailed investigation of each considered study. Therefore, to obtain the most objective sample with the minimized proportion of unsuitable articles, we developed a search query strategy that provided as complete and accurate result as possible.

At the first stage, we have highlighted the issues that we would like to address:Which types of machine learning are the most used in medicine? Are there different preferences for them in the different areas of medicine? If there are, then why?Which data types are the most used in medicine for artificial intelligence algorithms? How are the different data types treated?How the artificial intelligence methods used to diagnose the illnesses that are the most common causes of death?

Based on these questions, we selected four categories of data:Medical area (MA), or the field of medicine, in which the AI-based methods are used (we used pulmonology, gastroenterology, orthopedics, reproductive medicine, neurology, and cardiology);Cases (C)—the specific diseases that are considered as the leading causes of death (we used COVID-19, cancer, cardiac ischemia, stroke, and diabetes);Learning methods (SupL, UnsupL, DL)—types of machine learning (supervised, unsupervised, and deep learning, respectively);Data types (DT), that are commonly used in medicine as input for artificial intelligence algorithm (time series, images, health parameters).

To each of these categories, we assigned a set of keywords, the union of which describes each of the categories in the most complete way. For instance, the category *supervised learning* contains the names of the specific algorithms belonging to this type of machine learning, such as support vector machines, decision trees, linear regression, etc. The full transcripts of all used categories are shown in the Table in Appendix A. Figure 2 shows the scheme of the query construction.

We can highlight two types of queries. The first one is shown in Figure 2A. By this approach we have selected the articles that meet following the requirement: to use the particular data type to feed an artificial intelligence method and related to the particular medical area or the case—the specific disease (scheme 1 and 2 on the Figure 2A, respectively). Here, the entity *Artificial intelligence* is a unification of all considered sets of methods, i.e., unsupervised, supervised and deep learning. All keywords corresponding to the each category were united by the logical **OR.**

The queries of the second type were constructed according to the scheme on Figure 2B. Here, each set of the entity *Artificial intelligence* was used separately to narrow the search results down to the particular learning methods. Here, each query gave as a result the intersection of one of the learning methods of interest **AND** one of the other three sets. Here, each operand of the logical **AND** was a unification of the keywords using the logical **OR**.

The papers with paper types “Review”, “Systematic Review”, and “Meta-analysis” were excluded from the final sample using PubMed search filters. After collecting the number of articles found with each query, we selected the most prominent results and features, and performed the more particular review to establish the causes for these features on the example of specific cases.

All of the queries processed in this paper are valid for 5 February 2023.

### 2.2. The Co-Occurrence Network Analysis

The co-occurrence network (see Figure 3) was created based on keywords from 10,000 papers found in PubMed by search query consisting of all keywords from SupL, UnsupL, and DL sets of data using VOSviewer version 1.6.18 (Centre for Science and Technology Studies, Leiden University, The Netherlands). VOS stands for visualization of similarities and provides a mapping technique used for reconstruction of bibliometric maps [38]. In our research, we used network visualization that colors the items according to the cluster to which they belong. Clusters, or communities, are detected in VOSviewer using algorithm based on the modularity function [39]. For more particular analyses, we selected the largest items in each cluster and analyzed its within-cluster connections, as well as the most strong connections with items from other clusters, in order to provide the interpretation of obtained co-occurrence pattern of each chosen item in the context of its closest surrounding.

## 3. Results and Discussion

### 3.1. Network Analysis

The VOSviewer identified five clusters, which can be easily interpreted based on the keywords they contain (Figure 3). First, a **signal processing cluster** (purple lines) includes such areas of application as *electroencephalography* and *electrocardiography*. Second, we highlight a **deep learning cluster** (blue lines) which includes deep learning approaches, e.g., *neural networks, computer*, *deep learning*, and *artificial intelligence*. Third, a **machine learning cluster** (red lines) includes classical *machine learning* techniques, e.g., *support vector machine*, *principal component analysis*, etc. Fourth, an **image processing cluster** (yellow lines) includes *convolutional neural networks* as a main tool and a *magnetic resonance imaging* as a main area of application. Fifth, a **retrospective studies cluster** (green lines) mainly considers the *reproducibility of results*, *prognosis*, and methods for the evaluation of algorithms performance, e.g., *ROC curve*, etc. In the rest part of this section, we give an interpretation of the most evident within and between cluster structures.

The *deep learning* being the largest item of the deep learning cluster has strong within-cluster connections with the famous deep learning methods (neural networks and convolutional neural networks) (Figure 4A). Furthermore, presence of an *artificial intelligence* in this cluster suggests that medical studies usually associate an artificial intelligence with deep learning algorithms. Second, the deep learning cluster includes *COVID-19* and *SARS-CoV-2*, indicating that deep learning applications in medicine were influenced by COVID-19 pandemic that might have spark the development of medical image analysis using artificial intelligence methods. This assumption can be supported by the presence of *tomography, X-ray computed*, and *lung neoplasms* in the same cluster. The *deep learning* item has various connections with other clusters. The strongest connections are observed with the machine learning cluster, image processing cluster, and the retrospective studies cluster. Connections with image processing cluster indicate a wide application of the deep learning methods for analysis of neurological images. Connections with the retrospective studies cluster can also be interpreted as COVID-related because a lot of retrospective studies of COVID-19 aftermaths were published since 2020, particularly with applications of the artificial intelligence methods. Moreover, the connections with the retrospective studies mean the wide application of deep learning methods for disease prediction and prognosis (e.g., lung neoplasms, breast neoplasms, etc.).

The *retrospective studies* cluster contains such nodes as *ROC curve*, *prognosis*, *reproducibility of results*, *diagnosis*, *predictive value of tests*, *image interpretation*, and *sensitivity and specificity* (Figure 4B). Most likely, these studies include development of the decision support systems and automated systems for medical monitoring, diagnosis, and prognosis. These fields of application require interpretability of algorithms and reproducibility of results which explains the presence of the *reproducibility of results* and evaluation metrics, e.g., ROC curve, area under curve, sensitivity, and specificity. Considering between-cluster links, we observe strong connections with machine learning and deep learning clusters, indicating using both traditional machine learning and deep learning algorithms. Finally, there is a link to *magnetic resonance imaging* which, together with connections with *tomography, X-ray computed*, and *breast neoplasms*, may indicate retrospective studies preformed the context of diseases’ effects on human health, including the long-term effects of disease and epidemiological impact. Machine learning methods were used to make predictions of risks in patients with COVID-19 [41,42,43], as well as other diseases and conditions [44,45].

The *machine learning* node has within-cluster connections with the most traditional machine learning algorithms, such as support vector machine, decision trees, principal component analysis, logistic models, etc. (Figure 4C). Considering between-cluster links, we highlight the strongest connection with the deep learning algorithms confirming an obvious association between machine learning and deep learning algorithms. In addition, machine learning clusters have other important between-cluster links, including image processing, retrospective studies, and electroencephalography. In the retrospective studies, machine learning reaches such nodes as prognosis and diagnosis, suggesting the use of ML algorithms in these areas, and ROC curve, area under curve (AUC), reproducibility, sensitivity, and specificity. Again, using ML in diagnostics and prognosis raises questions of algorithms’ performance, their evaluation, and the reproducibility of results. The remaining between-cluster links illustrate areas of ML application for the different types of data. Thus, there are links to image processing and magnetic resonances imaging belonging to the image processing cluster, and a link to electroencephalography representing a signal processing cluster. Note, that another important type of data, electronic health records, is connected to the machine-learning node via a within-cluster link. Therefore, we conclude that machine learning finds its application in the three major types of medical data.

*Electroencephalography* and *electrocardiography* are the nodes of the *signal processing cluster*, they represent very important signals describing human state (Figure 4D,E). Considering the structure of the links outgoing from these nodes, we found that both data types have connections with the nodes of the deep learning cluster, including neural networks and artificial intelligence. This is probably due to the recurrent neural networks, a deep learning algorithm for signal processing. Second, they both link with the different nodes of the machine learning cluster, including principal component analysis, support vector machine, and clusterization. The main difference between electroencephalography and electrocardiography is that the first has a link to the *brain*, which is obvious, and that electroencephalography links to convolutional neural network. This difference arises from the nature of the signals. Unlike electrocardiography, electroencephalography includes multichannel recordings from the electrodes having certain spatial locations. Therefore, its analysis usually implies considering spatial distributions of the signal features, e.g., spectral amplitude across the spatial 2-D dimension [46]. The second aspect is that electroencephalography analysis often relies on the time-frequency decomposition that results in the 2-D distribution of the spectral power across frequency bands and time samples. Finally, there are 1-D convolution neural networks that are used for the analysis of such signals [47].

Taken together, we report the following insights from the data. First, medical applications consider deep learning as a part of machine learning that includes deep neural networks and artificial intelligence. Second, deep learning finds its application in image processing mostly, while traditional machine learning algorithms usually work with signals and health records. Third, using deep learning and machine learning methods raises the question of reproducibility. Fourth, prognosis and diagnosis mostly rely on traditional machine learning algorithms rather than deep learning methods because they pay much attention to the interpretability and stability of the algorithms.

### 3.2. Queries Results

In this subsection, we describe the search query results. Figure 5 shows the results of the queries complied following Figure 2B, with Figure 5; a build using queries 5, 8, and 11, Figure 5B—query 1, and Figure 5C—queries 3, 6, and 9. Generally, Figure 5 highlights the dominance of the *time series* as the data type used to feed artificial intelligence method. The *supervised learning* appears to be the most common choice in medical research. This figure further shows that most articles satisfying queries 5, 8, and 11 from Figure 2B investigate the combination of medical time series and supervised learning (Figure 5A). *Imaging* is the second most used data type in AI-based medical studies, predominantly used in neurology and in combination with supervised learning methods. The *health parameters* are also widely used as a data type, mostly required in reproductive medicine and neurological studies (Figure 5B). In the next sections, we consider the different cases of using supervised/unsupervised/deep learning methods for the different types of medical data more thoroughly.

The main idea of **unsupervised learning** is the categorization of the data without the labels provided for the training algorithm. A classic task for the classification of unlabeled data is clustering or segmentation of a dataset into groups based upon patterns extracted in the process of self-supervised learning.

Figure 5A demonstrates that the considerable attention of medical researchers is given to unsupervised machine learning with time series. The brief review of literature revealed the great importance of clusterization task in various areas of medicine. In reproductive medicine, this combination is used to divide the fetal biological signals (ECG, MEG, EEG) from mothers’ [48,49,50,51,52], as well as for extraction of anomalies and significant features. In particular, PCA was applied for fetal heart activity removal from MEG [53], to estimate the iron status of term newborns [54], and detection of fetal brain activity of different modalities [55,56]. Such studies lie on the interception of reproductive medicine and neurology, which shows dominance in both use of time series for AI-based methods (Figure 5B), and unsupervised methods application (Figure 5C).

In obstetrics, unsupervised learning solves the crucial task of prediction of the labor outcomes. Clustering techniques proved their effectiveness for identifying condition clusters associated with preterm birth [57], birth weight prediction [58], and detection of opening and closing of the fetal cardiac valves [59].

Our results demonstrated a notable amount of neurology studies with clusterization methods for detection of different brain patterns on imagery data (Figure 5C). Bayesian non-parametric regression was applied to detect brain regions based on stimulus response-related activity [60]. In addition, segmentation of brain tissues is a vital for detection of neurological diseases and regions of atrophies [61,62,63].

Considering the particular cases, the application of unsupervised machine learning is rather moderate (Figure 5C). However, we can notice the considerable amount of literature dedicated to the unsupervised learning in diabetes and COVID-19 studies (Figure 6A, based on queries 4, 7, and 10 from Figure 2B).

Our results demonstrated that the **supervised learning** is the most popular choice for the medical purposes. In particular, a bunch of literature is dedicated to the combination of supervised learning and time series (Figure 5A).

In reproductive medicine, supervised machine learning is applied for prediction of anomalies and evaluation of risks (Figure 5C). Such methods as logistic regression, decision tree model, naive Bayes classification, support vector machine, random forest algorithm, and stochastic gradient boosting method are used for preeclampsia detection [64,65,66]. The Decision Forest demonstrated the most promising results for fetal anomaly status prediction [67]. Other literature are dedicated to the prediction of Caesarean section risk factors [68,69,70,71,72]. Supervised learning in combination with electronic health records demonstrated a high effectiveness in prediction of postpartum depression [73,74,75,76], and in combination with resting state fMRI-based functional connectivity demonstrated a high performance in the classification of major depressive disorder [77,78].

In addition, our results reveal a dominance of supervised machine learning applications in the stroke and diabetes studies (Figure 6A). A brief review shows that prediction and prognosis become a goal of research in the both cases. In stroke studies, predictions of outcomes, course of the disease and evaluation of risks are crucial for intervention strategy planning [79,80]. Similarly, supervised learning methods were used for assessment of diabetes risks based on identified key variables [81,82], as well as for prediction of risks in patients with diabetes [83,84].

Our search results demonstrate that, in general, the **deep learning** is rather rare in medical studies. The main reason is that the artificial neural networks, representing most of this category, leave almost no room for interpretation of the results [85], and this particular outcome of artificial intelligence in medicine is crucial. At the same time, deep learning-based methods are a popular choice in heart disease studies, and particular ways of their application vary from one research to another (Figure 6A). In Ref. [86], authors describe a method for coronary artery disease (CAD) detection using a deep convolutional neural network based on facial photos. Another group of studies uses ECG signals as input data for deep learning algorithms to detect abnormalities associated with CAD [87]. However, in the context of CAD diagnosis, prediction, and risk evaluation, the most popular choice of input dataset is various techniques of cardiac imaging, such as cardiac CT and MRS [88].

Figure 6B (based on query 2 from Figure 2A) shows that the specific **data types** prevail in studies associated with particular cases. For instance, AI-based studies of cardiac ischemia mostly use time series, whereas COVID-19 studies rely on imaging techniques, and AI-based methods use health parameters as data in stroke studies. This result may provide insight into the particular goals of research on each disease. In particular, cardiac-related AI applications often focus on fast and accurate identification of pathological activity on ECG [89,90,91]. In the case of COVID-19, not only identification but also localization of lesions on CT scan becomes a goal for AI-based research [92,93]. In stroke studies, algorithms use health parameters for risk evaluations and outcome predictions [94,95,96].

## 4. Conclusions

In this paper, we analyzed area-specific and data-specific domains of AI methods applications in modern medicine using a data mining approach. We found in PubMed a large collection of papers that met the specially chosen keywords and applied the two types of analysis to them: the co-occurrence network analysis and statistical analysis of query results. These data analysis approaches complemented each other. We obtained the following key results:Deep learning finds its application in image processing mostly (especially in cardiology for heart disease studies based on analysis of cardiac imaging data), while traditional machine learning algorithms usually work with signals and health records. In general, deep learning methods are rather rare in medical studies because they leave almost no room for interpretation of the results.Using deep learning and machine learning methods raises the question of reproducibility.Prognosis and diagnosis mostly rely on traditional machine learning algorithms rather than deep learning methods because they pay much attention to the interpretability and stability of the algorithms.The considerable attention of medical researchers is given to unsupervised machine learning with time series (ECG, EEG, MEG, etc.), especially to clusterization and segmentation techniques, regression, and PCA methods in various areas of medicine (reproductive medicine, neurology, cardiology, COVID-19, diabetes studies, etc.).Supervised machine learning is the most popular choice for medical purposes: a bunch of literature is dedicated to the combination of supervised machine learning methods and time series. Methods such as logistic regression, decision tree model, naive Bayes classification, support vector machine, random forest algorithm, and stochastic gradient boosting method are used for the prediction of anomalies and evaluation of risks in reproductive medicine, neurology, stroke, diabetes studies, etc.The specific data types prevail in studies associated with particular cases. For instance, AI-based studies of cardiac ischemia mostly use time series, whereas COVID-19 studies rely on imaging techniques, and AI-based methods use health parameters as data in stroke studies.

However, this research has certain limitations. First of all, we acknowledge the lack of thorough analysis of compliance of each paper with the corresponding topic of interest. In this research, we considered the samples of a large amount of papers obtained from PubMed database, which was impossible to manually analyze. However, we consider our search query strategy specifically designed to provide the most accurate and conclusive results as a “rejection criteria” based on the inclusion of certain keywords into the title or text of the article. Of course, this method cannot guarantee the absence of unsuitable articles in the final samples, but we can ensure that their number is relatively small and did not affect the presented results.

We believe that the data systematized in this work will support the specialist in choosing the most appropriate method for the task he/she faces related to the analysis of medical data of a certain type in a given medical area. In this way, the paper presented may serve as a kind of handbook.

## Figures and Tables

**Figure 1 ijerph-20-05335-f001:**
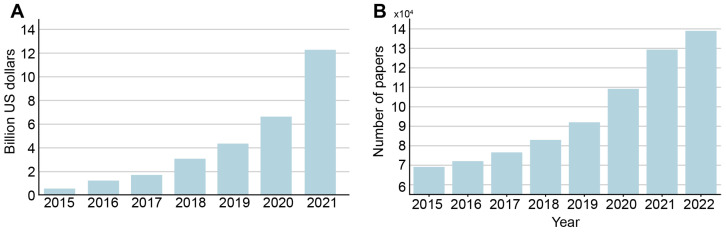
(**A**) dynamics of venture capital investment in artificial intelligence systems for medicine and healthcare, according to CB Insights, USD billion. Based on data from ‘State of AI 2021 Report’ [Internet]. Source: https://www.cbinsights.com/research/report/ai-trends-2021/ (accessed on 21 September 2022). (**B**) dynamics of number of AI in medicine papers by year indexed in the PubMed database.

**Figure 2 ijerph-20-05335-f002:**
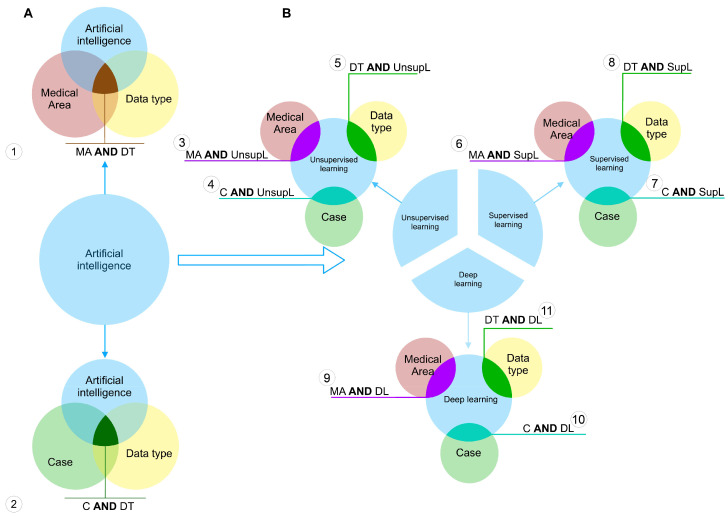
Query construction scheme. (**A**)–the construction scheme of queries of the first type. (**B**)–the construction scheme of queries of the second type.

**Figure 3 ijerph-20-05335-f003:**
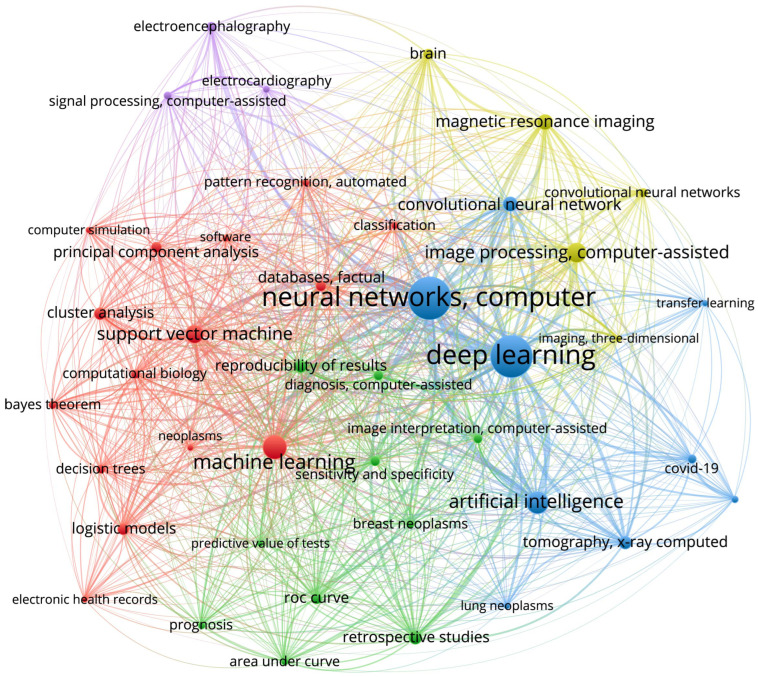
The co-occurrence network constructed using VOSviewer. The size of the nodes are determined by the weight of the corresponding item that indicate the importance of the item, and the color is determined by the cluster to which the item belongs. The methodology of VOSviewer visualization technique is provided in details in [40].

**Figure 4 ijerph-20-05335-f004:**
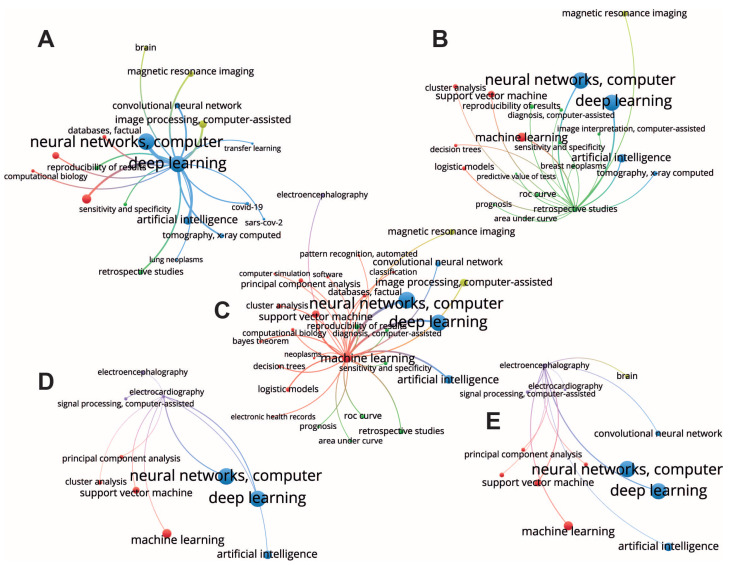
Selected subnetworks of interest with the largest nodes corresponding to deep learning (**A**), retrospective studies (**B**), machine learning (**C**), electrcardiography (**D**) and electroencephalography (**E**).

**Figure 5 ijerph-20-05335-f005:**
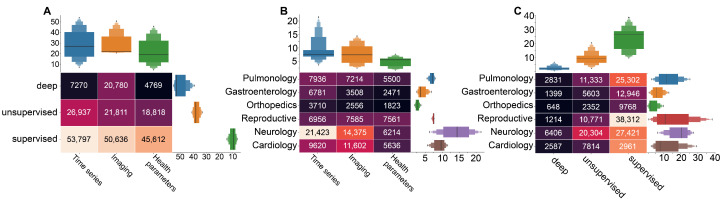
Results of query research of the PubMed database with the number of papers acquired via the corresponding search queries. (**A**) the use of machine learning types in combination with different medical data types; (**B**) medical data in artificial intelligence research in different medical areas; and (**C**) the use of machine learning methods in different medical area.

**Figure 6 ijerph-20-05335-f006:**
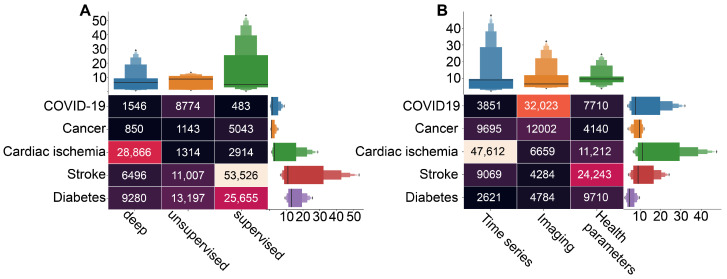
The results of the queries with the number of the acquired papers on particular diseases research in combination with different data (**A**) and machine learning types (**B**).

## Data Availability

Data are contained within the article.

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
