# Peer review of "Analysis of Publication Activity and Research Trends in the Field of AI Medical Applications: Network Approach"

_ijerph, 2023, doi:10.3390/ijerph20075335_

Round 1

Reviewer 1 Report

1. List the main contributions clearly in the Introduction.

2. Make sure to state the total number of the included papers, their paper types, and their temporal trend analysis in the text.

3. I suggest analysing your paper's strengths and limitations in the discussion section.

Author Response

1. List the main contributions clearly in the Introduction.

Response:

We thank the Reviewer for this valuable remark. We rephrased the contribution in the Introduction section (lines 123-132).

2. Make sure to state the total number of the included papers, their paper types, and their temporal trend analysis in the text.

Response:

We thank Reviewer for this comment. We added the figure with temporal trend in the text of the paper (Figure 1B). As to the total number of the included papers, we present the number of papers acquired via search queries for each category of interest on the corresponding figures (Figure 5 and Figure 6). Finally, we excluded the paper types «Review», «Systematic Review» and «Meta-Analysis» from each sample. We added this remark to the Methods section of our manuscript (lines 178-179).

3. I suggest analysing your paper's strengths and limitations in the discussion section.

Response:

We thank Reviewer for this comment. We added the statement to the Conclusion section (lines 397-405).

Reviewer 2 Report

The authors propose: Analysis of publication activity and research trends in the field of AI medical applications: network approach. I have some concerns and my suggestions are listed below:

1- The contribution is not adequately explained in the abstract. There is no driving force behind the essay. The information was not presented in a way that was understandable and straightforward. The main idea of the work should be emphasized in the abstract section.

2- The writers should focus on the study's main issue in the introduction and provide a Literature Review in the form of tables to identify research gaps and innovations.

3- It is crucial to enhance experimental findings, validate them, and compare them to MORE different approaches. More discussions and analyses are necessary.

4- The benefits and drawbacks of the connected works were not assessed by the authors. Please assess how their research differs from other studies in the section OF related work. What do they possess that others lack? Why or how are they superior? What is novel or new in this?

5- It is crucial to describe the data mining approach’ computational complexity.

6- For experiments, nonparametric tests should be used.

7- The authors should be clear about the method's benefits and drawbacks. What are the methodology(ies) and limitation(s) used in this work? Please list benefits to daily life and go over any research limitations.

8- If you have developed any code or software, it is recommended that you provide a link to the code for other readers and to enhance the impact of the paper and its applicability

Author Response

The authors propose: Analysis of publication activity and research trends in the field of AI medical applications: network approach. I have some concerns and my suggestions are listed below:

1- The contribution is not adequately explained in the abstract. There is no driving force behind the essay. The information was not presented in a way that was understandable and straightforward. The main idea of the work should be emphasized in the abstract section.

Response:

We thank the Reviewer for the criticism. We have rewritten the abstract in accordance with the comments. The revised part of the abstract is highlighted in blue

2- The writers should focus on the study's main issue in the introduction and provide a Literature Review in the form of tables to identify research gaps and innovations.

Response:

We thank the Reviewer for this valuable remark. We added a paragraph in the Introduction explaining the potential research gaps (lines 105-112). Also, we added the main contributions in the Introduction section (lines 120-129).

3- It is crucial to enhance experimental findings, validate them, and compare them to MORE different approaches. More discussions and analyses are necessary. 

Response:

Our paper aims at the exploration of the AI-based medicine research using network approach and analysis of existing trends based on PubMed. Our findings are based on the results of PubMed search queries and analysis of the number of papers obtained by the different search queries. Our goal was to explore how are the AI-based methods used in healthcare research, which approaches and techniques are the most popular, and to discuss the potential reasoning behind the obtained results.

4- The benefits and drawbacks of the connected works were not assessed by the authors. Please assess how their research differs from other studies in the section OF related work. What do they possess that others lack? Why or how are they superior? What is novel or new in this?

 Response:

The goal of our paper was to highlight the tendencies of AI-based methods development in medical research. With this goal, we collected the co-occurrence information from PubMed database and analyzed the number of papers on different topics and their intersection (used methods, medical data type, medical areas, etc). For discussion, we tried to substantiate each result we found by brief review of some of the articles in each sample.

5- It is crucial to describe the data mining approachcomputational complexity.

 Response:

Our data mining approach was to collect information about the number of articles published in PubMed database. Using analysis of the co-occurrence network constructed with the VOSviewer software, we detected the main clusters of interest in AI-based healthcare research. Then, we proceeded with the thorough analysis of publication activity in various categories of medical AI research, including research on different AI-based methods applied to different types of medical data. We analyzed the results of query processing in the PubMed database over the past 5 years obtained via a specifically designed strategy for generating search queries based on the thorough selection of keywords from different categories of interest.

6- For experiments, nonparametric tests should be used.

Response: 

In our manuscript, we analyzed the number of AI-related medical papers obtained from PubMed database with search queries generated by our designed algorithm. In this case, the statistical methods are not applicable.

7- The authors should be clear about the method's benefits and drawbacks. What are the methodology(ies) and limitation(s) used in this work? Please list benefits to daily life and go over any research limitations.

 Response:

We thank reviewer for this valuable suggestions. We added the following statement to the Conclusion section (lines 397-405):

«However, this research has certain limitations. First of all, we acknowledge the lack of thorough analysis of compliance of each paper with the corresponding topic of interest. In this research, we considered the samples of a large amount of papers obtained from PubMed database, which was impossible to manually analyze. However, we consider our search query specifically designed to provide the most accurate and conclusive results as a "rejection criteria" based on the inclusion of certain keywords into the title or text of the article. Of course, this method cannot guarantee the absence of unsuitable articles in the final samples, but we can ensure that their number is relatively small and did not affect the presented results».

8- If you have developed any code or software, it is recommended that you provide a link to the code for other readers and to enhance the impact of the paper and its applicability

Response:

We agree on importance of sharing the code used to obtain scientific results. We have described the method of generating the search queries in detail and provided all the keywords used for queries in a table format in the Appendix.

Reviewer 3 Report

I had a chance to review this paper entitled "Analysis of publication activity and research trends in the field of AI medical applications: network approach," and I found a few objections that must be rectified before any acceptance.

1. The abstract should be improved with proper situating of the problem area, objectives, and findings.

2. You Need to add contributions in the 2nd last paragraph in the form of bullets.

3. second, third and fourth paragraphs in the introduction section are without any references. You cannot write any para without any reference.

3. In the methodology section, the role of AI in diagnostics or CAD should be defined and explained with citations.

4. The colour of nodes (blue and red) and their sizes are not defined in Figure 3.

5.  Author can add a table format for discussing recent and relevant publications and their comparisons after the introduction section.

Author Response

I had a chance to review this paper entitled "Analysis of publication activity and research trends in the field of AI medical applications: network approach," and I found a few objections that must be rectified before any acceptance.

1. The abstract should be improved with proper situating of the problem area, objectives, and findings.

Response:

We thank the reviewer for this comment. We have rewritten the abstract in accordance with the remarks.

2. You Need to add contributions in the 2nd last paragraph in the form of bullets.

Response:

We thank the reviewer for the criticism. We added the paragraph that explains the contributions more clearly to the introduction section (lines 123-132)

3. second, third and fourth paragraphs in the introduction section are without any references. You cannot write any para without any reference.

Response:

We backed up the mentioned paragraphs with references.

4. In the methodology section, the role of AI in diagnostics or CAD should be defined and explained with citations.

Response:

We added the paragraph addressing the potential research gaps related to use of AI in medical research, including the existing problems with CDSS (lines 107-115).

5. The colour of nodes (blue and red) and their sizes are not defined in Figure 3.

Response:

We thank the reviewer for this remark. We have added the description of the colour and size of the nodes in the caption of Figure 3 with the reference.

5. Author can add a table format for discussing recent and relevant publications and their comparisons after the introduction section.

Response:

In our research, we analyzed not the particular papers, but the tendencies and directions of AI development in healthcare and medicine based on the analysis of large amount of papers obtained from PubMed database with search queries. The method to generate the search queries is a feature of this research, designed thoroughly and specifically to obtain most conclusive and adequate results for each topic of interest (area of medicine, use of different AI-based methods in combination with medical data of different modalities, etc.)